# ON THE INTERACTION BETWEEN SUPERVISION AND SELF-PLAY IN EMERGENT COMMUNICATION

**Ryan Lowe,**[*] **Abhinav Gupta**[*]
MILA

**Jakob Foerster, Douwe Kiela**
Facebook AI Research

**Joelle Pineau**
Facebook AI Research
MILA

## ABSTRACT

A promising approach for teaching artificial agents to use natural language involves using human-in-the-loop training. However, recent work suggests that current machine learning methods are too data inefficient to be trained in this way from scratch. In this paper, we investigate the relationship between two categories of learning signals with the ultimate goal of improving sample efficiency: imitating human language data via supervised learning, and maximizing reward in a simulated multi-agent environment via self-play (as done in emergent communication), and introduce the term *supervised self-play (S2P)* for algorithms using both of these signals. We find that first training agents via supervised learning on human data followed by self-play outperforms the converse, suggesting that it is not beneficial to emerge languages from scratch. We then empirically investigate various S2P schedules that begin with supervised learning in two environments: a Lewis signaling game with symbolic inputs, and an image-based referential game with natural language descriptions. Lastly, we introduce population based approaches to S2P, which further improves the performance over single-agent methods.[1]

## 1 INTRODUCTION

Language is one of the most important aspects of human intelligence; it allows humans to coordinate and share knowledge with each other. It is also crucial for human-machine interaction, as human language is a natural means by which to exchange information, give feedback, and specify goals. A promising approach for training agents to solve problems with natural language is to have a "human in the loop", meaning we collect problem-specific data from humans interacting directly with our agents for learning. However, human-in-the-loop data is expensive and time-consuming to obtain as it requires continuously collecting human data as the agent's policy improves, and recent work suggests that current machine learning methods (e.g. from deep reinforcement learning) are too data-inefficient to be trained in this way from scratch (Chevalier-Boisvert et al., 2019). Thus, an important open problem is: how can we make human-in-the-loop training as data efficient as possible?

To maximize data efficiency, it is important to fully leverage all available training signals. In this paper, we study two categories of such training methods: imitating human data via supervised learning, and self-play to maximize reward in a multi-agent environment, both of which provide rich signals for endowing agents with language-using capabilities. However, these are potentially competing objectives, as maximizing environmental reward can lead to the resulting communication protocol drifting from natural language (Lewis et al., 2017; Lee et al., 2019). The crucial question, then, is how do we best combine self-play and supervised updates? This question has received surprisingly little attention from the emergent communication literature, where the question of how to bridge the gap from emergent protocols to natural language is generally left for future work (Mordatch & Abbeel, 2018; Lazaridou et al., 2018; Cao et al., 2018).

---

[*]These two authors contributed equally. Work done primarily while RL was at Facebook AI.
Correspondence to: `ryan.lowe@cs.mcgill.ca`, `abhinavg@nyu.edu`

[1]Code is available at `https://github.com/backpropper/s2p`.

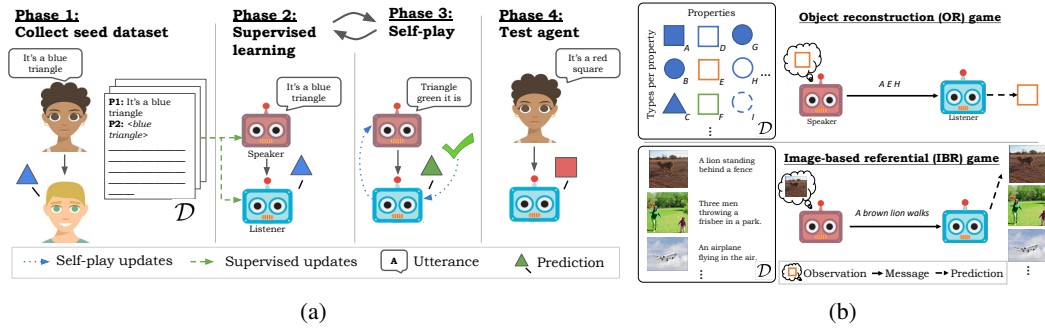

Figure 1: (a) Diagram of the supervised self-play (S2P) procedure (phases 1-3) and the testing procedure considered in this work (phase 4). (b) The environments considered in this paper (Sec. 4).

Our goal in this paper is to investigate algorithms for combining supervised learning with self-play — which we call supervised self-play (S2P) algorithms — using two classic emergent communication tasks: a Lewis signaling game with symbolic inputs, and a more complicated image-based referential game with natural language descriptions. Our first finding is that supervised learning followed by self-play outperforms emergent communication with supervised fine-tuning in these environments, and we provide three reasons for why this is the case. We then empirically investigate several supervised-first S2P methods in our environments. Existing approaches in this area have used various ad-hoc schedules for alternating between the two kinds of updates (Lazaridou et al., 2017), but to our knowledge there has been no systematic study that has compared these approaches. Lastly, we propose the use of population-based methods for S2P, and find that it leads to improved performance in the more challenging image-based referential game. Our findings highlight the need for further work in combining supervised learning and self-play to develop more sample-efficient language learners.

## 2 RELATED WORK

In the past few years, there has been a renewed interest in the field of emergent communication (Sukhbaatar et al., 2016; Foerster et al., 2016; Lazaridou et al., 2017; Havrylov & Titov, 2017) culminating in 3 NeurIPS workshops. Empirical studies have showed that agents can autonomously evolve a communication protocol using discrete symbols when deployed in a multi-agent environment which helps them to play a cooperative or competitive game (Singh et al., 2019; Cao et al., 2018; Choi et al., 2018; Resnick* et al., 2019; Evtimova et al., 2018).

While the idea of promoting coordination among agents through communication sounds promising, recent experiments (Lowe et al., 2019; Chaabouni et al., 2019; Kottur et al., 2017; Jaques et al., 2019) have emphasized the difficulty in learning meaningful emergent communication protocols even with centralized training.

Apart from the above advances in emergent communication, there has been a long outstanding goal of learning intelligent conversational agents to be able to interact with humans. This involves training the artificial agents in a way so that they achieve high scores while solving the task and their language is interpretable by humans or close to natural language. Recent works also add another axis orthogonal to communication where the agent also takes a discrete action in an interactive environment (de Vries et al., 2018; Mul et al., 2019). Lewis et al. (2017) introduced a negotiation task which involves learning linguistic and reasoning skills. They train models imitating human utterances using supervised learning and found that the model generated human-like captions but were poor negotiators. So they perform self-play with these pretrained agents in an interleaved manner and found that the performance improved drastically while avoiding language drift. Lee et al. (2019) also propose using an auxiliary task for grounding the communication to counter language drift. They use visual grounding to learn the semantics of the language while still generating messages that are close to English.

A recent trend on using population based training for multi-agent communication is a promising avenue for research using inspirations from language evolution literature (Smith et al., 2003; Kirby,

2014; Raviv & Arnon, 2018). Cultural transmission is one such technique which focuses on the structure and compression of languages, since a language must be used and learned by all individuals of the culture in which it resides and at the same time be suitable for a variety of tasks. Harding Graesser et al. (2019) shows the emergence of linguistic phenomena when a pool of agents contact each other giving rise to novel creole languages. Li & Bowling (2019); Cogswell et al. (2019); Tieleman et al. (2018) have also tried different ways of imposing cultural pressures on agents, by simulating a large population of them and pairing agents to solve a cooperative game with communication. They train the agent against a sampled *generation* of agents where the *generation* corresponds to the different languages of the different agent at different times in the history.

Our work is inspired from these works where we aim to formalize the recent advancements in using self-play in dialog modeling, through the lens of emergent communication.

## 3 METHODS

### 3.1 PROBLEM DEFINITION

Our agents are embedded in a multi-agent environment with $N$ agents where they receive observations $o \in O$ (which are functions of a hidden state $S$) and perform actions $a \in A$. Some actions $A_L \subset A$ involve sending a message $m \in A_L$ over a *discrete*, *costless* communication channel (i.e. a *cheap talk* channel (Farrell & Rabin, 1996)). The agents are rewarded with a reward $r \in R$ for their performance in the environment. We assume throughout that the environment is cooperative and thus the agents are trained to maximize the sum of rewards $R = \sum_{t=1:T} \sum_{i=1:N} r_{i,t}$ across both agents. This can be thought of as a cooperative partially-observable Markov game (Littman (1994)).

We define a target language $L^* \in \mathcal{L}$, usually corresponding to natural language, that we want our agents to learn (we further assume $L^*$ can be used to achieve high task reward). In this paper, we consider a language $L \in \mathcal{L}$ to be simply a set of valid messages $A_L$ and a mapping between observations and messages in the environment, $L : O \times A_L \mapsto [0, 1]$. For example, in an English image-based referential game (Section 4) this corresponds to the mapping between images and image descriptions in English. We are given a dataset $\mathcal{D}$ consisting of $|\mathcal{D}|$ (observation, action) pairs, corresponding to $N_e$ 'experts' (for us, $N_e = 2$) playing the game using the target language $L^*$. Our goal is to train agents to achieve a high reward in the game while speaking language $L^*$ with an 'expert'. Specifically, we want our agents to *generalize* and to perform well on examples that are not contained in $\mathcal{D}$.

To summarize, we want agents that can perform well on a collaborative task with English-speaking humans, and we can train them using a supervised dataset $\mathcal{D}$ and via self-play.

### 3.2 SUPERVISED SELF-PLAY (S2P)

In recent years, there have been several approaches to language learning that have combined supervised or imitation learning with self-play. In this paper, we propose an umbrella term for these algorithms called *supervised self-play* (S2P). S2P requires two things: (1) a multi-agent environment where at least one agent can send messages over a dedicated communication channel, along with a reward function that measures how well the agents are doing at some task; and (2) a supervised dataset $\mathcal{D}$ of experts acting and speaking language $L^*$ in the environment (such that they perform well on the task). Given these ingredients, we define S2P below (see Figure 2).

**Definition 3.1. Supervised self-play (S2P).** *Supervised self-play is a class of language learning algorithms that combines: (1) self-play updates in a multi-agent language environment, and (2) supervised updates on an expert dataset $\mathcal{D}$.*

S2P algorithms can differ in how they combine self-play and supervised learning updates on $\mathcal{D}$. When supervised learning is performed before self-play, we refer to the dataset $\mathcal{D}$ as the *seed data*. Why might we want to train our agents via self-play? Won't their language diverge from $L^*$? One way to intuitively understand why S2P is beneficial is to think in terms of constraints. In our set-up, there are two known constraints on the target language $L^*$: (1) it is consistent with the samples from the supervised dataset $\mathcal{D}$, and (2) $L^*$ can be used to obtain a high reward in the environment. Thus, finding $L^*$ can be loosely viewed as a constrained optimization problem, and enforcing both constraints should clearly lead to better performance.

### 3.3 ALGORITHMS FOR S2P

Here we describe several methods for S2P training. Our goal is not to exhaustively enumerate all possible optimization strategies, but rather provide a categorization of some well-known ways to combine self-play and supervised learning. To help describe these methods, we further split the seed dataset $\mathcal{D}$ into $\mathcal{D}_{train}$, which is used for training, and $\mathcal{D}_{val}$ which is used for early-stopping. We also visualize the schedules in Figure 2.

**Emergent communication with supervised fine-tuning (`sp2sup`):** We first perform self-play updates until the learning converges on the task performance. It is then followed by supervised updates on $\mathcal{D}_{train}$ until the listener performance converges on the dataset $\mathcal{D}_{val}$.

**Supervised learning with self-play (`sup2sp`):** This is the complement of the above method which involves doing supervised updates until convergence on $\mathcal{D}_{val}$ followed by self-play updates until convergence on the task performance.

**Random updates (`rand`):** This is the method used in (Lazaridou et al., 2017). At each time step, we sample a bernoulli random variable $z \sim Bernoulli(q)$ where $q$ is fixed. If $z = 1$, we perform one supervised update, else we do one self-play update, and repeat until both losses convergence on $\mathcal{D}_{val}$.

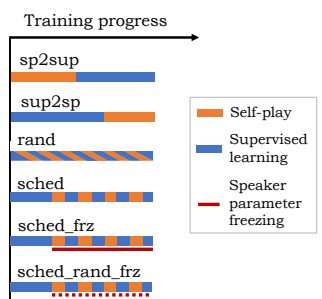

Figure 2: A visual representation of the different S2P methods.

**Scheduled updates (`sched`):** We first pretrain the listener and the speaker until convergence on $\mathcal{D}_{val}$. Then we create a *schedule*, where we perform $l$ self-play updates followed by $m$ supervised updates, and repeat until convergence on the dataset.

**Scheduled updates with speaker freezing (`sched_frz`):** This method is based on the findings of Lewis et al. (2017), who do `sched` S2P while freezing the parameters of the speaker during self-play to reduce the amount of language drift. In our case, we freeze the parameters of the speaker after the initial supervised learning.

**Scheduled updates with random speaker freezing (`sched_rand_frz`):** Experimentally, we noticed that `sched_frz` didn't perform well in self-play. Thus, we introduce a variation, we sample a bernoulli random variable $z \sim Bernoulli(r)$ where $r$ is fixed. If $z = 1$, we freeze the parameters of the speaker during both self-play and supervised learning, else we allow updates to the speaker as well.

### 3.4 POPULATION-BASED S2P (POP-S2P)

As explained above, the goal of S2P is to produce agents that follow dataset $\mathcal{D}$ while maximizing reward in the environment. However, there are many such policies satisfying these criteria. This results in a large space of possible solutions, that increases as the environment grows more complex (but decreases with increasing $|\mathcal{D}|$). Experimentally, we find that this can result in diverse agent policies. We show this in Figure 3 by training 50 randomly initialized agents on the image-based referential game (defined in Sec. 4) the agents can often make diverse predictions for a given image (Figure 3a) and achieve variable performance when playing with other populations with a slight preference towards their own partner (the diagonal in Figure 3b).

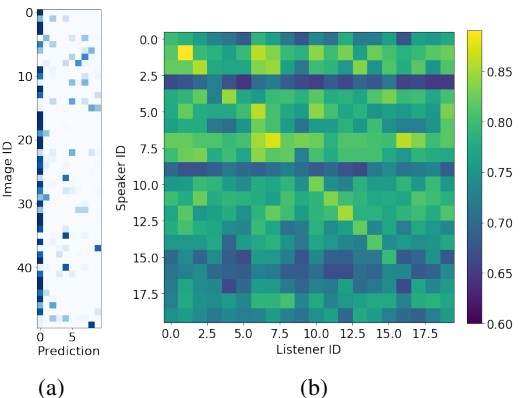

Figure 3: Results from training 50 S2P agents on the IBR game with $|\mathcal{D}| = 10000$. (a) The agents have a range of predictions on many images. (b) When playing with each other, the agents exhibit uneven performance (color is mean reward, yellow is higher), indicating policy variability.

Inspired by these findings, we propose to augment S2P by training a *population* of $N$ agents, and subsequently aggregating them back into a

single agent (the 'student'). We call this population-based S2P (Pop-S2P). While there are many feasible ways of doing this, in this paper we train the populations by simply randomizing the initial seed, and we aggregate the populations using a simple form of policy distillation (Rusu et al., 2016). Another simple technique to boost performance is via ensembling where we simply take the majority prediction at each time step.

## 4 ENVIRONMENTS & IMPLEMENTATION DETAILS

We consider environments based on classical problems in emergent communication. These environments are cooperative and involve the interaction between a speaker, who makes an observation and sends a message, and a listener, who observes the message and makes a prediction (see Figure 1b). Our goal is to train a listener such that it achieves high reward when playing with an expert speaking the target language $L^*$ on inputs unseen during training.[2]

**Environment 1: Object Reconstruction (OR)**   Our first game is a Lewis signaling game (Lewis, 1969) and a simpler version of the Task & Talk game from Kottur et al. (2017), with a single turn and a much larger input space. The speaker agent observes an object with a certain set of properties, and must describe the object to the listener using a sequence of words. The listener then attempts to reconstruct the object. More specifically, the input space consists of $p$ *properties* (e.g. shape, color) of $t$ *types* each (e.g. triangle, square). The speaker observes a symbolic representation of the input, consisting of the concatenation of $p = 6$ one-hot vectors, each of length $t = 10$. The number of possible inputs scales as $t^p$. We define the vocabulary size (length of each one-hot vector sent from the speaker) as $|V| = 60$, and the number of words (fixed length message) sent to be $T = 6$.

For our target language $L^*$ for this task, we programatically generate a perfectly compositional language, by assigning each object a unique word. In other words, to describe a 'blue shaded triangle', we create a language where the output description would be "blue, triangle, shaded", in some arbitrary order. By 'unique symbol', we mean that no two properties are assigned the same word. The speaker and listener policies are parameterized using a 2-layer linear network (results were similar with added non-linearity and significantly worse with 1-layer linear networks) with 200 hidden units. During both supervised learning and self-play, the listener is trained to minimize the cross-entropy loss over property predictions.

**Environment 2: Image-Based Referential game with natural language (IBR)**   Our second game is the communication task introduced in Lee et al. (2018). The speaker observes a target image $d^*$, and must describe the image using a set of words. The listener observes the target image along with $D$ distractor images sampled uniformly at random from the training set (for us, $D = 9$), and the message $y_{d^*}$ from the speaker, and is rewarded for correctly selecting the target image. For this game, the target language $L^*$ is English — we obtain English image descriptions using caption data from MS COCO and Flickr30k. We set the vocabulary size $|V| = 100$, and filter out any descriptions that contain more than 30% unknown tokens while keeping the maximum message length $T$ to 15.

Similar to (Mordatch & Abbeel, 2018; Sukhbaatar et al., 2016), we train our agents end-to-end with backpropagation. Since the speaker sends discrete messages, we use the Straight-Through version of Gumbel-Softmax (Jang et al., 2017; Maddison et al., 2017) to allow gradient flow to the speaker during self-play ($\mathcal{J}_{\text{self-play}}$). The speaker's predictions are trained on the ground truth English captions $m^*$ using the cross entropy loss $\mathcal{J}_{\text{spk-supervised}}$. The listener is trained using the cross-entropy loss $\mathcal{J}_{\text{lsn-supervised}}$ where the logits are the reciprocal of the mean squared error which was found to perform better than directly minimizing MSE loss in Lee et al. (2018). The mean squared error is taken over the listener's image representation $b_{lsn}$ of the distractor (or target) image and the message representation given as input. The loss functions are defined as:

$$\mathcal{J}_{\text{spk-supervised}}(d^*) = -\sum_{t=1}^{T} \log p_{spk}(m_t|m_{<t}, d^*)$$

$$\mathcal{J}_{\text{lsn-supervised}}(m^*, d^*, D) = -\sum_{d=1}^{D+1} \log(\texttt{softmax}(1/p_{lsn}(m^*) - b_{\text{lsn}}(d))^2)$$

---

[2]Our approach could equally be used to train a speaker of language $L^*$; we leave this to future work.

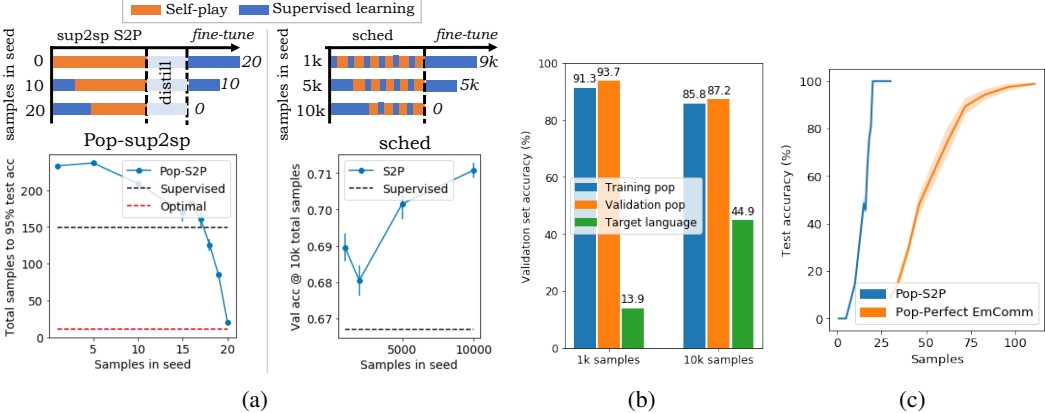

Figure 4: (a) Left: In the OR game, best performance (number of total samples required to achieve 95% test accuracy, lower is better) for S2P is achieved when all of the samples are in the seed. 0 on the x-axis corresponds to `sp2sup` and Optimal is the actual (minimum) number of samples required to solve this optimization problem (see Appenix B). Right: This is also the case in the IBR game, where performance is measured by the generalization accuracy using 10k total training samples (higher is better). (b) Adding more samples to initial supervised learning in the IBR game improves agents' generalization to $L^*$. (c) Even when we learn the perfect distribution with emergent communication in the OR game, it still performs worse than Pop-S2P (using `sup2sp` S2P).

$$\mathcal{J}_{\text{self-play}}(d^*, D) = -\sum_{d=1}^{D+1} \log(\texttt{softmax}(1/p_{lsn}(y_{d^*}) - b_{lsn}(d))^2)$$

where $y_{d^*}$ is the concatenation of $T$ one-hot vectors $y_{d^*}^t = \texttt{ST-GumbelSoftmax}(p_{spk}^t)$.

We use the same architecture as described in Lee et al. (2018). The speaker and listener are parameterized by recurrent policies, both using an embedding layer of size 256 followed by a GRU (Cho et al., 2014) of size 512. We provide further hyperparameter details in Table 1 in the Appendix.

## 5 DO SUPERVISED LEARNING BEFORE SELF-PLAY

A central question in our work is how to combine supervised and self-play updates for effective pre-training of conversational agents. In this section, we study this question by conducting experiments with two schedules: training with emergent communication followed by supervised learning (`sp2sup`), and training with supervised learning followed by self-play (`sup2sp`). We also interpolate between these two regimes by performing the `rand` and `sched` on $0 < n < |\mathcal{D}|$ samples, followed by supervised fine-tuning on the remaining $|\mathcal{D}| - n$ samples.

Our first finding is that it is best to use all of your samples for supervised learning before doing self-play. This can be seen in Figure 4: when all of the samples are used first for supervised learning, the number of total samples required to solve the OR game drastically, and in the IBR game the accuracy for a fixed number of samples is maximized (Figure 4a). While this may seem to be common sense, it in fact runs counter to the prevailing wisdom in some emergent communication literature, where languages are emerged from scratch with the ultimate goal of translating them to natural language.

To better understand why it is best to do supervised learning first, we now conduct a set of targeted experiments using the environments from Section 4. Results of our experiments suggest three main explanations:

**(1) Emerging a language is hard.** *For many environments, with emergent communication it's often hard to find an equilibrium where the agents meaningfully communicate.* The difficulty of 'emergent language discovery' has been well-known in emergent communication (Lowe et al., 2017), so we will only briefly discuss it here. In short, to discover a useful communication protocol agents

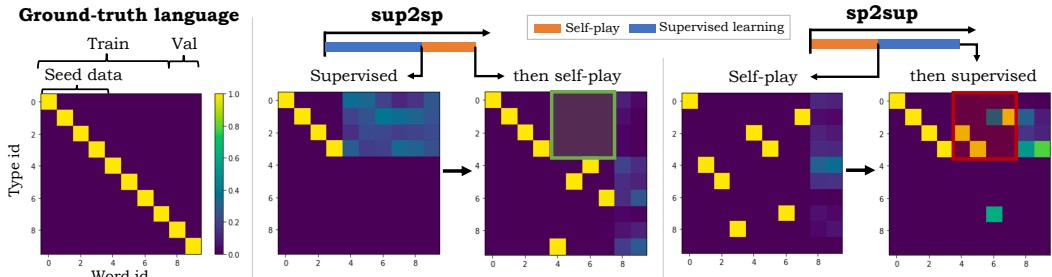

Figure 5: Results from the OR game with 1 property and 10 types. When the supervised updates are performed first (supervised data available for words $0 - 3$), then the self-play updates make sensible predictions for the unknown words $4 - 7$. When the self-play updates are performed first, the subsequent supervised updates merely correct the predictions for words $1 - 4$, without enforcing the constraint that each word should result in a separate type to solve the task.

have to coordinate repeatedly over time, which is difficult when agents are randomly initialized, particularly in environments with sparse reward. Compounding the difficulty is that, if neither agent communicates and both agents act optimally given their lack of knowledge, they converge to a Nash equilibrium called *babbling equilibrium* (Farrell & Rabin, 1996). This equilibrium must be overcome to learn a useful communication protocol. In S2P, the initial language supervision can help overcome the discovery problem, as it provides an initial policy for how agents could usefully communicate (Lewis et al., 2017).

**(2) Emergent languages are different than natural language.** *Even if one does find an equilibrium where agents communicate and perform well on the task, the distribution of languages they find will usually be very different from natural language.* This is a problem because, if the languages obtained through self-play are sufficiently different from $L^*$, they will not be helpful for learning. This is seen for the OR game in Figure 4a, where 17 samples are required in the seed before S2P outperforms the supervised learning baseline. We speculate that this is due to the different pressures exerted during the emergence of artificial languages and human languages.

Thankfully, we can learn languages closer to $L^*$ by simply adding more samples to our initial supervised learning phase. We show this in Figure 4b, where we train populations of 50 agents on the IBR game and use Pop-S2P to produce a single distilled agent. With both 1K and 10K initial supervised samples, the distill agent generalizes to agents in the validation set of their population. However, the distilled agent trained with 10000 samples performs significantly better when playing with an expert agent speaking $L^*$, indicating that the training agents from that population speak languages closer to $L^*$.

**(3) Starting with self-play violates constraints.** *Even if you have 'perfect emergent communication' that learns a distribution over languages under which $L^*$ has high probability, current methods of supervised fine-tuning do not properly learn from this distribution.* What if we had all the correct learning pressures, such that we emerged a distribution over languages $\mathcal{L}$ with structure identical to $L^*$, and then trained a Pop-S2P agent using this distribution? Surprisingly, we find that S2P with all of the samples in the seed performs better than even this optimistic case, in terms of providing useful information for training a Pop-S2P agent. We conduct an experiment in the OR game where we programmatically define a distribution over compositional languages $\mathcal{L}_c$, of which our target language $L^*$ is a sample. Each language $L \in \mathcal{L}_c$ has the same structure and are obtained by randomly permuting the mapping between the word IDs and the corresponding type IDs, along with the order of properties in an utterance. Next, we compare two distilled policies using 50 populations: one is distilled from S2P populations (trained with $X$ samples), and the other is distilled from 'perfect emergent communication' and fine-tuned on $X$ samples. As can be seen in Figure 4c, we show that when we train a Pop-S2P agent on 50 of these compositional populations, we still need $3X$ more samples than regular Pop-S2P (trained on 50 S2P agents with all of the samples in the seed) to reach 95% test accuracy[3].

---

[3]Here the value of $X$ is the corresponding number of samples on the horizontal axis in Figure 4c.

To understand why this happens, we conduct a case study in an even simpler setting: single-agent S2P in the OR game with $p = 1, t = 10, |V| = 10$. We find that agents trained via emergent communication consistently learn to solve this task. However, as shown in Figure 5, when subsequently trained via supervised learning on $\mathcal{D}$ to learn $L^*$, the learned language is no longer coherent (it maps different words to the same type) and doesn't solve the task. On the other hand, agents trained first with supervised learning are able to learn a language that both solves the task and is consistent with $\mathcal{D}$.

Intuitively, what's happening is that the samples in $\mathcal{D}$ are also valid for solving the task, since we assume agents speaking $L^*$ can solve the task. Thus, self-play after supervised learning simply 'fills in the gaps' for examples not in $\mathcal{D}$.[4] Emergent languages that start with self-play, on the other hand, contain input-output mappings that are inconsistent with $L^*$, which must be un-learned during subsequent supervised learning.

In theory, the above issue could be resolved using Pop-S2P; if the distilled agent could use the population of emergent languages to discover structural rules (e.g. discovering that the languages in the OR game in Figure 4c are compositional), it could use the samples from $\mathcal{D}$ to refine a posterior distribution over target languages that is consistent with these rules (e.g. learning the distribution of compositional languages consistent with $\mathcal{D}$). Current approaches to supervised fine-tuning in language, though, do not do this (Lazaridou et al., 2017; Lewis et al., 2017). An interesting direction for future work is examining how to apply Bayesian techniques to S2P.

# 6 EXPLORING VARIANTS OF S2P

## 6.1 POPULATION-BASED S2P

In this section, we aim to show that (1) S2P outperforms the supervised learning baseline, and (2) Pop-S2P outperforms S2P. We conduct our experiments in the more complex IBR game, since the agents must communicate in English, and measure performance by calculating the accuracy at different (fixed) numbers of samples. Our baseline is then the performance of a supervised learner on a fixed number of samples.

We show the results in Figure 6. We first note that, when both 1k and 10k samples are used for supervised learning, S2P (`sched`) outperforms the supervised learning baseline. We can also see that the population-based approach outperforms single agent S2P (`sched`) by a significant margin. We also compare our distillation method to an ensembling method that keeps all 50 populations at test time, and find that ensembling performs significantly better, although it is much less efficient. This suggests that there is room to push distilled Pop-S2P to even better performance.

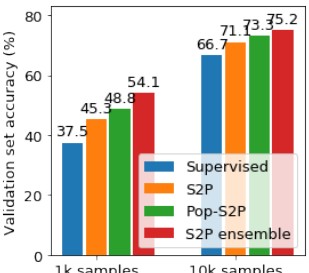

Figure 6: S2P (`sched`) outperforms the supervised baseline in the IBR game, and is in turn outperformed by Pop-S2P.

## 6.2 EXAMINING S2P SCHEDULES

In this section, we aim to: (1) evaluate several S2P schedules empirically on the IBR game; and (2) attain a better understanding of S2P through quantitative experiments.

**Parameter freezing improves S2P** We show the results comparing different S2P schedules in Figure 7a. We find that in this more complex game, the `sup2sp` S2P performs much worse than the other options. We also see that adding freezing slightly improves the performance on the target language (Figure 8 in the Appendix also shows that it converges more quickly). We hypothesize that this is because it reduces the language drift that is experienced during each round of self-play updates (Lee et al., 2019). Overall, however, the difference between different S2P schedules is relatively small, and it's unclear if the same ordering will hold in a different domain.

---

[4]In practice, we find that self-play updates can undo some of the learning of $\mathcal{D}$, which is why we apply an alternating schedule.

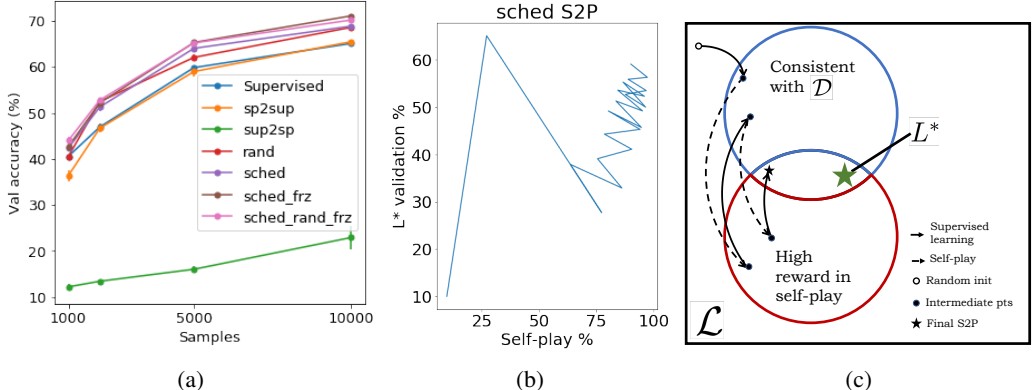

(a)          (b)          (c)

Figure 7: (a) Comparing test performances of different S2P methods on the IBR game. For each method, we picked the model that gave the best performance on $\mathcal{D}_{val}$. (b) 2D visualization of S2P (`sched`) performance over the course of training, in terms of performance on $L^*$ (vertical axis) and performance in self-play (horizontal axis). The zig-zag patterns indicates that most self-play updates result in a short-term decrease in target language performance. (c) Visualization of the role of the supervised and self-play updates in `sched` S2P.

**Self-play acts as a regularizer**    What is the role of self-play in S2P? We can start to decipher this by taking a closer look at the `sched` S2P. We plot the training performance of this method in Figure 7b. Interestingly, we notice from the zig-zag pattern that the validation performance usually goes *down* after every set of self-play updates. However, the overall validation performance goes up after the next round of supervised updates. This is also reflected in the poor performance of the `sup2sp` S2P in Figure 6.

This phenomenon can be explained by framing self-play as a form of regularization: alternating between supervised and self-play updates is a way to satisfy the parallel constraints of 'is consistent with the dataset $\mathcal{D}$' and 'performs well on the task'. We visualize this pictorially in Figure 7b: while a set of self-play updates results in poor performance on $\mathcal{D}$, eventually the learned language moves closer to satisfying both constraints.

## 7   DISCUSSION

In this work, we investigated the research question of how to combine supervised and self-play updates, with a focus on training agents to learn a language. However, this research question is not only important for language learning; it is also a important in equilibrium selection and learning social conventions (Lerer & Peysakhovich, 2019) in general games. For example, in robotics there may be a trade-off between performing a task well (moving an object to a certain place) and having your policy be interpretable by humans (so that they will not stumble over you). Examining how to combine supervised and self-play updates in these settings is an exciting direction for future work.

There are several axes of complexity not addressed in our environments and problem set-up. First, we consider only single-state environments, and agents don't have to make temporally extended decisions. Second, we do not consider pre-training on large text corpora that are separate from the desired task (Radford et al., 2019; Devlin et al., 2018). Third, we limit our exploration of self-play to the multi-agent setting, which is not the case in works such as instruction following (Andreas & Klein, 2015). Introducing these elements may result in additional practical considerations for S2P learning, which we leave for future work. Our goal in this paper is not to determine the best method of S2P in all of these settings, but rather to inspire others to use the framing of 'supervised self-play algorithms' to make progress on sample efficient human-in-the-loop language learning.

ACKNOWLEDGEMENTS

We are very grateful to Angeliki Lazaridou, with whom discussions at ICML 2019 and her simultaneous work (Lazaridou et al., 2020) shifted the direction of this work considerably. We also thank Jean Harb, Liam Fedus, Amy Zhang, Evgeny Naumov, Cinjon Resnick, Igor Mordatch, and others at MILA and Facebook AI Research for discussions related to the ideas in this paper. Special thanks to Arthur Szlam and Kavya Srinet for discussing their ongoing work with us. RL is supported in part by a Vanier Scholarship.

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

## A  HYPERPARAMETERS

We provide hyperparameter details in Table 1.

| Hyperparameter | Values |
|---|---|
| Learning rate | 1e-2, 1e-3, 2e-3, 6e-3, 1e-4, 5e-4, 6e-4 |
| Model architecture | Linear, Bilinear, Non-Linear |
| Number of encoders (perfect emcomm) | 1, 2, 5, 10, 20, 50, 100, 200, 500, 1000 |
| Hidden layer size (Linear) | 200, 500, 1000 |
| Number of encoders (Pop-S2P) | 20, 40, 50, 60, 80, 100 |
| Number of distractors | 1, 4, 9 |
| GRU hidden size | 256 |
| Word embedding size | 512 |
| Image embedding size (from pretrained Resnet50) | 2048 |
| Batch size | 1, 512, 1000 |
| Random seeds | 0, 1, 2, 3, 4 |
| Optimizer | Adam, SGD |
| Dropout | 0, 0.3 |
| Gumbel relaxation temperature | 1 |
| Vocabulary size | 100, 200, 500, 1000, 5000 |
| Max sentence length | 12, 15, 20, 30, 50 |
| $m$ in `sched` | 0, 1, 30, 40, 50, 70 |
| $l$ in `sched` | 0, 30, 40, 50 |
| $q$ in `rand` | 0.75 |
| $r$ in `sched_rand_frz` | 0.5 |
| Number of initial supervised steps (pretraining) | 0, 1000, 2000, 3000, 5000 |

Table 1: Hyperparameters considered in S2P training.

## B  CALCULATION OF OPTIMAL SAMPLE COMPLEXITY IN OR GAME

Here we provide a quick calculation for how quickly a human might learn a new compositional language $L$ in the OR game in as few examples as possible, which we use as a baseline in Figure 4a. We assume a OR game with $p = 6$ properties, $t = 10$ types, $T = 6$ words sent per message (concatenated together), and $|V| = 60$ vocabulary size. If this language $L$ is compositional, then each word in the vocabulary is assigned to 1 type. Thus, we need to learn 60 total assignments. In this analysis we assume we can construct (i.e. hand-design) the samples seen by the human, and thus the final number should be considered something like a lower bound.

Since $T = 6$, we get information about 6 word←type assignments for every sample. However, this information is entangled as we don't know which word corresponded to which type. Thus, we (1) divide the problem up by first constructing 9 (word sequence, object) sample pairs where none of the object types overlap between each sample. With this information, we are able to narrow down the word←type assignments into 10 groups of 6 (that is, in each group we have 6 words corresponding to 6 types, but we don't know which type belongs to which word). Note we don't need 10 samples as the last one can be inferred by exclusion. (2) We then construct 5 more samples where each type belongs to a separate group. We can do this because $t > p$. Because each type belongs to a separate group, cross-referencing the words observed from samples in (1) and (2) uniquely defines each word←type assignment. Note again we don't need 6 samples as the last one can be inferred by exclusion. This gives us a total of $9 + 5 = 14$ samples.

## C  ADDITIONAL PLOTS

We show training curves for various S2P schedules.

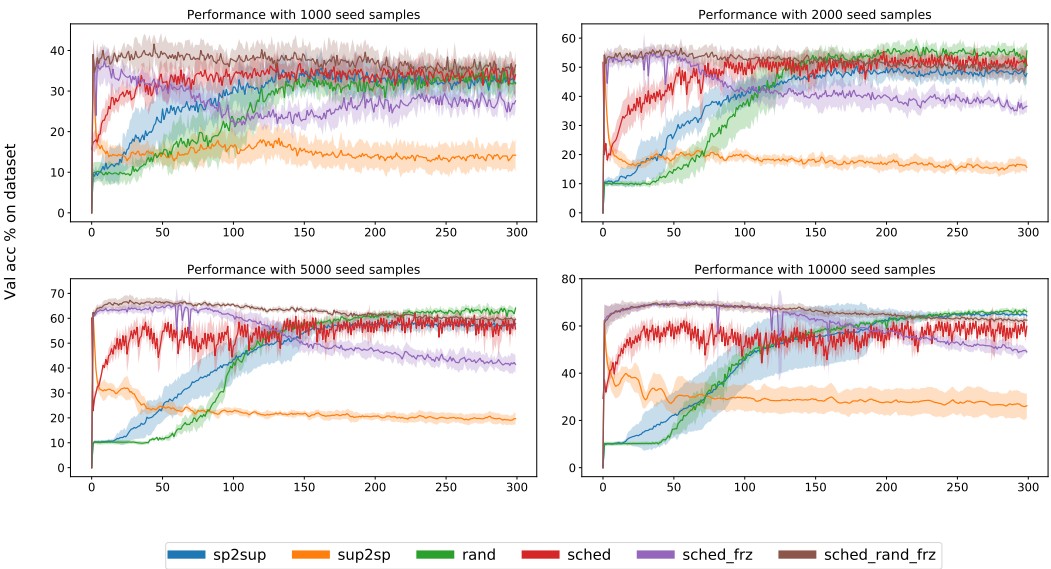

Figure 8: Training curves for various S2P methods in the IBR game described in §4.

