# OpenReview forum: "On the interaction between supervision and self-play in emergent communication"
_ICLR.cc/2020/Conference — Accept (Poster)_

### Official Review · AnonReviewer3 · 2019-10-23
**Official Blind Review #3**

**Rating:** 6

**Review:**

This paper explores the effect of ordering supervised learning and self-play on the resultant language learnt between agents. The topic is of high relevance to the ICLR community and makes several interesting insights useful to anyone learning control of a multi-agent system where communication amongst agents is applicable. I have several suggestions for improvements below, but all I believe are feasible to make within the time period of the rebuttal with the most necessary being:

1) The naming of methods in Section 5 is not consistent with those introduced in section 3.3. For example, in the first paragraph ec2supervised is presumably sp2sup and sched is presumably sup2sp? Similarly, on page 7 (S2P and Pop-S2P) and Figure 4. Please revise and ensure consistency throughout.

2) Figures 4b, 6 and 7 only present single values. Are these average values from repeated runs? If so please quantify variance.

3) The conclusion in Section 7 at the bottom of page 8 that "S2P performs much worse than the other options" is contrary to previous results. Can the authors please comment on what features of the environment caused this difference?

4) Appendix A includes details of hyperparameters, but some details remain unclear. Specifically, hyperparameter ranges swept over are shown but how were they then chosen from? Are they optimised for each environment and algorithm? What does the bold text in the table represent? If it is chosen values, why do only some parameters have chosen values? These are important details to enable reproduction of the paper.

Minor Comments:
In Section 3.3, if all these methods are "well known ways to combine self-play and supervised learning" can all be supported by an (or preferably multiple) exemplar publications that used these method previously. Directly linking each to the previous work will further clarify the contribution this specific paper makes and help readers new to the area gain insight across the multiple papers this work builds upon.

Figure 3b, colour is representative of performance. Is this mean accumulated reward? Please clarify to increase how informative this visualisation is, as currently it is unclear if yellow or blue is the desired value.

Page 5, small typo "introduced in the Lee et al. (2017)" should be "introduced in Lee et al. (2017)".

Figure 4b, the legend is blocking two bars and their corresponding value. It looks like moving to the bottom right may help, or placing above the plot.

Figure 4 caption refers to a subfigure (d) that is not included.

On Page 6, the reference to babbling equilibrium should include a citation for interested readers to learn more about this well established concept.

On Page 7 there is a reference to Figure r4b, is this intended to be a reference to Figure 4b right?

Figure 6 appears after Figure 7. Maintaining ordered numbering would be preferable.

On Page 9 it is noted some experimental results are in the Appendix but as there is a page and a half of space remaining before the 10 page limit, I would encourage to include all results in the main body of the paper.

Multiple references do not list a publication venue (e.g. Evtimova et al., Lazaridou et al. 2018, Tieleman et al. 2018) or cite Arxiv versions when the work has been later published (e.g. Jacques et al. 2018 was published at ICML 2018).

Figure 9 caption should state the environment.

**Experience Assessment:**

I have published one or two papers in this area.

**Review Assessment: Checking Correctness Of Derivations And Theory:**

N/A

**Review Assessment: Checking Correctness Of Experiments:**

I carefully checked the experiments.

**Review Assessment: Thoroughness In Paper Reading:**

I read the paper thoroughly.

---

> ### Author Response · Authors · 2019-11-11
> **Response to Reviewer #3**
>
> We thank the reviewer for the kind and insightful review.
>
> > Naming not consistent
>
> Yes, we indeed mean ec2supervised as sp2sup and sched as sup2sp in the first paragraph of Section 5. Thanks for pointing this out, we will correct all the naming errors and ensure consistency throughout in the final version.
>
> > Adding error bars
>
> Yes, the current values are the mean over runs from 5 different seeds. We will add error bars to show variance in the final version.
>
> > The conclusion in Section 7 that ‘S2P performs much worse than the other options’ is contrary to previous results.
>
> In this Section, we are referring to the “poor performance of the *sup2sp* method”. The sup2sp method is defined in Figure 2 and Section 3.3, and refers to one of the many instantiations of the (more general) S2P framework. So what we are saying is not that S2P performs worse, but that sup2sp performs worse than other methods of S2P. This is shown in Figure 7a (not Figure 6, as we mistakenly wrote in the text). The reason for this is that, since self-play is a form of regularizer (see Figure 7b), doing self-play updates often leads to worse performance on the task (until more supervised updates are done). So, if you finish training with self-play updates without performing supervised updates, your performance will be quite low. We will clarify this in the final version.
>
> > Add more hyperparameter details
>
> Thank you for pointing this out, we will indeed update and clarify this in the final version.
>
> > Figure 3b, colour is representative of performance. Is this mean accumulated reward? Please clarify to increase how informative this visualisation is, as currently it is unclear if yellow or blue is the desired value.
>
> Yes, it shows the performance (mean accumulated reward) over 50 pairs of speakers and listeners, so a higher value (yellow) is desirable. It shows that the performance of both the speakers and listeners when paired randomly is found to be quite variable, although we do observe a slight preference towards their own partner (yellow diagonal). Due to space constraints, we chose to omit this but we will clarify this in the final version.
>
> > if all these methods are "well known ways to combine self-play and supervised learning" can all be supported by an (or preferably multiple) exemplar publications that used these method previously.
>
> We do refer to Lewis et al. 2017 published at EMNLP’17 (oral) and Lazaridou et al. 2016 published at ICLR’17 (oral) for two S2P methods, sched_frz and rand respectively. The sup2sp and sp2sup are the baseline models which we use for comparing other sophisticated approaches, and which we compare to show that supervised learning before self-play generally improves performance (Section 5). The sched_rand_frz is a novel extension to the sched_frz method which we found was more stable and sometimes performed slightly better.
>
> > Reference errors, fig refs/naming errors
> Thanks for pointing out these, we will add references to the published versions of these papers and fix the references/naming to the figures in the text.

---

### Official Review · AnonReviewer2 · 2019-10-23
**Official Blind Review #2**

**Rating:** 8

**Review:**


Summary
---

(motivation)
To develop language speaking agents we can teach them to mimic human language
or to solve tasks that require communication. The latter is efficient, but
the former enables interpretability. Thus we combine the two in an attempt
to take advantage of both advantages. This paper studies a variety of ways to
combine these approaches to inform future work that needs to make this tradeoff.

(approach)
The trade-off is studied using reference games between a speaker and a
listener. Goal oriented _self-play_ and human _supervision_ are considered two contraints one
can put on a network during learning. This work considers algorithms that vary
when self-play and supervision are used (e.g., training with self-play then supervision,
or supervision then self-play, or alternating back and forth between the two).
Additional variations freeze the speaker or distill an ensemble of agents into one agent.

(experiments)
A synthetic Object Reference game (OR) and a Image-Base Reference game (IBR) with real images are used for evaluation. Performance is accuracy at image/object guessing.
1. (OR) Like previous work, this work finds that emergent languages are imperfect at supporting their goals and cannot be understood by agents that only understand a human language like English.
2. (OR) Pre-training with supervision then fine-tuning with self-play is superior to pre-training with self-play then fine-tuning with supervision. This is presented as surprising from the perspective of language emergence literature, which is though of as pre-training with self-play.
3. (IBR) Distilling an agent from an ensemble of 50 independently trained agents outperforms training single agents from scratch, but is still not as good as the whole ensemble.

Self-play vs supervision schedules:
4. (IBR) Supervision (using image captions) followed by self-play performs much worse than all other approaches.
5. (IBR) Alternating between supervision and self play (e.g., randomly choosing supervision or self-play every iteration) performs best.



Strengths
---

The curricula considered by this paper seem to have a sigificant impact on performance. These are new and could be important for future work on language learning, which may have considered the sup2sp setting from figure 7a without considering the sched setting.

The diversity of experiments provided and the analysis help the reader get a better sense for how emergent communication models work.

It's nice to see experiments on both a toy setting and a setting with realistic images.

Future directions suggested throughout the paper are interesting.


Weaknesses
---


* The 3rd point of section 5 is presented as a major conclusion of this paper, but it is not very surprising and I don't see how it's very useful. The perspective of language emergence literature is presented a bit strangely. The self-play to supervision baseline seems to be presented as an approach from the language emergence literature. I don't think this is what any of that literature promotes exactly, though it is close. Generally, I (and likely others) don't think it's too surprising that trying to fine-tune a self-play model with language supervision data doesn't work very well, for the same reasons cited in this paper (point 3 of section 5). I think the general strategy when trying to gain practical benefits from self-play pre-training is a translation approach where the learned language is translated into a known language like English rather than trying to directly align it to English as does the supervision approach in this paper. This particular baseline would be more useful if the paper considered learning some kind of translation layer on top of the self-play pre-trained model.

* How significant are the performance differences in figure 7a, especially those between the frozen and non-frozen models? Is the frozen model really better or this performance difference just due to noise?

* I'm somewhat skeptical that these trends will generalize to other tasks/models. The main goal of this paper is to inform future work. That makes it even more important than normal that the trends identified here are likely to generalize well. Are these trends likely to generalize well? Does the paper address when these trends are expected to hold anywhere?


Minor Presentation Weaknesses:

* Figure 4: I think the sub-figures are mis-labeled in the caption.

* In the related work I'm not sure the concept of generations is right. I think it should refer to different languages of different agents across time rather than different languages of the same agent across time.


Missing details / clarification questions:

* What exactly does Figure 4c compare? Are both methods distilled from ensembles or is the blue line normal S2P while the other is distilled from an ensemble of compositional languages? It's not clear since point (3) in section 5 refers to the S2P result (not Pop-S2P) in that plot. I'm also assuming that PB-S2P means the same thing as Pop-S2P, but that's not made clear anywhere. Does PB stand for Population Based?

* In the rand setting how is convergence defined? Do both objectives need to converge or just one?

* In the sched_rand_frz setting what is r?

* In the IBR how are the distractor images picked?


Suggestions:

* Can't both self-play and supervision be used at the same time (just use a weighted combination of the two objectives)? I don't think the paper ever did this but it seems like a very useful variation to consider.


Preliminary Evaluation
---

Clarity: The writing is fairly clear, though some details are lacking.
Significance: This work could help inspire some future models in the language emergence literature.
Quality: Experiments are aligned with the paper's goals and support its conclusions.
Originality: The distillation approach and curricula are novel.

Overall the work could prove to be an interesting and useful reference point inside the language emergence literature so I recommend it for acceptance.



**Experience Assessment:**

I have published one or two papers in this area.

**Review Assessment: Checking Correctness Of Derivations And Theory:**

I assessed the sensibility of the derivations and theory.

**Review Assessment: Checking Correctness Of Experiments:**

I assessed the sensibility of the experiments.

**Review Assessment: Thoroughness In Paper Reading:**

I read the paper at least twice and used my best judgement in assessing the paper.

---

> ### Author Response · Authors · 2019-11-11
> **Response to Reviewer #2 (part 1/2)**
>
> We thank the reviewer for the kind and insightful review.
>
> > Point 3 of section 5 isn’t very surprising, since most emergent comm doesn’t do fine-tuning to transfer to human language. The point would be stronger if ‘translation layers’ were considered.
>
> In point 3 of section 5, we provide evidence for the claim that doing supervised learning before self-play is better than the converse. Indeed, while the question of how to bridge the gap from emergent communication to natural language is of significant interest to the emergent communication community (see e.g. the upcoming NeurIPS Workshop on ‘bridging the gap from emergent communication to natural language’ https://sites.google.com/view/emecom2019/), you are right that there is no consensus as to how this should be done, and very few papers (Lazaridou et al 2016, Havrylov et al 2017) have examined this explicitly (although there are many papers that examine how to make emergent language more compositional to be closer to human language). Two natural ways to do this are: (1) to fine-tune an emergent language using human data (similar to our single-agent sp2sup), or (2) to use a population of emergent languages as a kind of ‘prior’ that can then be fine-tuned with human data (our population-based sp2sup is inspired by this). You are correct that there is also some work (Andreas et al., 2017) on translating (e.g. computing an alignment) between the emergent language and natural language, and we think an interesting next step would be to compare the task performance of these models to various S2P schedules.
>
>
> > Unclear if the results in figure 7a are significant
>
> We will add error bars to the final version of our paper, which will clarify these concerns. Indeed, the difference between some of the schedules are not significant (we found that the standard error was in the range of 1-2%). Our analysis shows that there is not one S2P method that is clearly superior to all others, but slight improvements could be gained from switching schedules (which may be task dependent).
>
> > Skepticism about whether these trends will generalize to other tasks / models
>
> We are assuming that by ‘trends’, the reviewer is referring to the order of performance of different S2P schedules. Our belief is that there are some trends that will hold when moving to other tasks / models (the lower performance of sp2sup and sup2sp). However, as mentioned above, our analysis showed that while there is indeed some variation between different S2P methods, there is no method that is clearly superior to all others. We suspect that using a schedule (potentially with parameter freezing) will perform better than the random schedule, however this may be task-dependent. We will clarify this view in the paper.
>
> > Minor weaknesses
>
> Thank you for pointing these out, we will fix them in the final copy.

---

> ### Author Response · Authors · 2019-11-11
> **Response to Reviewer #2 (part 2/2)**
>
> Clarification questions
>
> > What exactly does Figure 4c compare? Are both methods distilled from ensembles or is the blue line normal S2P while the other is distilled from an ensemble of compositional languages? It's not clear since point (3) in section 5 refers to the S2P result (not Pop-S2P) in that plot. I'm also assuming that PB-S2P means the same thing as Pop-S2P, but that's not made clear anywhere. Does PB stand for Population Based?
>
> Figure 4c compares two distilled policies. One is distilled from S2P populations (trained with X samples), and one is distilled from ‘perfect emergent communication languages’ (defined in the text) and fine-tuned on X samples. So both are population-based. We apologize for the naming error, by PB-S2P we indeed mean Pop-S2P.
>
> > In the rand setting how is convergence defined? Do both objectives need to converge or just one?
>
> For the rand setting, both the objectives need to converge since we define convergence based on the performance of the listener on $\mathcal{D}_{val}$. Fig 9 in Appendix shows that they indeed converge after certain number of train steps.
>
> > In the sched_rand_frz setting what is r?
>
> We define r as the probability of freezing the speaker parameters as mentioned in Section 3.3. The actual number was mistakenly commented out in the submitted version. For reference, we use l=50 and m=50 for sched, r=0.5 for sched_rand_frz, and q=0.75 for rand. We will update this in the final version.
>
> > In the IBR how are the distractor images picked?
>
> They are picked using a uniform random distribution over all the images available in the dataset.
>
> > Can't both self-play and supervision be used at the same time (just use a weighted combination of the two objectives)? I don't think the paper ever did this but it seems like a very useful variation to consider.
>
> Yes this can indeed be done, by mixing gradient updates from both self-play and supervision in a single batch. This is quite close to the ‘random’ schedule (which alternates every example), and we don’t expect to see much difference, although it could indeed be tried.

---

### Official Review · AnonReviewer1 · 2019-10-24
**Official Blind Review #1**

**Rating:** 6

**Review:**

This paper investigated how two conflicting learning objectives; supervised and self-play updates could be combined with a focus on visual-grounded language tasks. With a different set of their combinations, the authors empirically found that alternating two learning updates may result in the best equilibrium state; consistency with samples in the supervised dataset and optimal state with high rewards in the task environment.

The paper is very well-written, and I really enjoyed reading it overall. There are some typos, presentation issues, and minor format issues (e.g., wrong naming) though. I do like this kind of simple but insightful result with enough empirical observations and discussions. Even though there is not that novel method proposed, the overall message found from the experiments, their interpretation by the authors, and meaningful comparisons to the past works in emergent communication are fair enough to learn high scientific values from it. The design of the experiment is again very simple (e.g., changing the size of data, switching two setups in different ways) but clear to understand. This work is a good example of how well-designed hypotheses and their empirical validation could contribute to the field. I also appreciate the large spectrum of literature surveys including from the recent advances (Lewis et al., 2017, Lee et al., 17) to the past literature in emergent communications such as Littman (1994) and (Farrell & Rabin, 1996).

One of my concerns is the lack of applications, especially on the tasks using more natural language. The two tasks; OR and IBR, seem to be very limited settings to evaluate how self-play operates with data supervision. As pointed out by the authors, supervision from the training data itself may include most of the unexplored cases of the task, leading a less chance to learn policies from the high rewards. I think more realistic tasks using natural language need to be considered: negotiation (e.g., Lewi’s task, “Decoupling strategy and generation in negotiation dialogues”), recommendation (e.g., “Recommendation as a Communication Game: Self-Supervised Bot-Play for Goal-oriented Dialogue”), and more. I agree with the point made by the authors that this work mainly focuses on investigation rather than exploitation. But, then it would be adding another emergent task where the self-play can learn many more policies than one in the supervised dataset.

Adding to the point, I was expecting to see non-task related metrics to measure the effectiveness of their appropriate combinations. For example, it would be better to add language-side metrics (e.g., perplexity, fluency, consistency) to measure how language degeneration varies by the different combinations. This issue is not addressed in the paper, and I guess this is mainly because of the limited usage of language in the two limited tasks. If the paper is only focusing on emergent language which is related to specific tasks, it would be better to tone-down a little bit and state the major difference of it with natural language.

The population-based S2P seems to be a bit incremental and unrelated to the main theme of the paper. To me, the motivation of adding POP into S2P based on the policy variability is somewhat different from the original claim about the combination of supervised and selfplay. Also, the improvements on IBR in Figure 7 are incremental, making the major claim of this work little divergent.

In terms of presentation, if you like to show how performance changes over the different sizes of data, it would be better to show it by graphs over different variations instead of the bar charts only with 10k and 50k sizes. In addition, the figures and captions need to be improved for better interpretation. I think they are written in a hurry or changed a lot in the last minutes. Please see some minor formatting issues below.


Minor comments:
Duplicate reference of (Lewis et al., 2017)
Some names defined in Section 3.3 and Section 5 are not exactly matched.
Figures and fonts in Figures 4 and 7 are a little difficult to understand. Especially, I can’t understand the two upper figures in Figure 4a
Captions in Figure 4 are not matched with the sub-figures.
Figure r4b -> Figure 4b


**Experience Assessment:**

I have published one or two papers in this area.

**Review Assessment: Checking Correctness Of Derivations And Theory:**

I assessed the sensibility of the derivations and theory.

**Review Assessment: Checking Correctness Of Experiments:**

I assessed the sensibility of the experiments.

**Review Assessment: Thoroughness In Paper Reading:**

I read the paper at least twice and used my best judgement in assessing the paper.

---

> ### Author Response · Authors · 2019-11-11
> **Response to Reviewer #1**
>
> Thank you for your kind and insightful review!
>
> > Considering other natural language tasks, e.g. negotiation or recommendation
>
> We agree that there are other tasks in NLP that have more direct applications than the OR and image-based referential (IBR) games. The reason we selected the IBR game is because it’s the most common game with natural language that has been used in previous work on emergent communication (e.g. [1, 2, 3]). Thus, it makes sense to compare various supervised and self-play schedules on this task. We agree that a strong step for future work would be expanding to other complex natural language tasks such as negotiation (note that the suggested ‘language recommendation’ paper came out on arXiv only two weeks before the ICLR submission deadline, and was accepted only after the deadline).
>
> > Adding non-task related metrics to study the language.
>
> This is an interesting suggestion. To help understand the difference in language generation policies for different S2P schedules, we will add qualitative examples to the Appendix, and investigate which other metrics we could add to compare the generated languages (most likely perplexity on the validation set, as it’s unclear how one would automatically measure ‘fluency’ and ‘consistency’). In our experience, adding self-play usually results in a decrease in perplexity (because you are adding an objective that’s not maximum likelihood), in exchange for better performance on the task.
>
> > Population-based S2P is incremental and unrelated
>
> Yes, we agree that introducing population-based S2P is somewhat orthogonal to the main theme of our paper. We think population-based approaches have a lot of potential for improving grounded language learning with self-play especially when language tasks become more complex. This is because, for more complex language tasks, we hypothesize that self-play will result in larger deviations from natural language, and population-based approaches can help alleviate this. While our current distilled Pop-S2P result doesn’t yet reach the ensemble result, the S2P ensemble results in Figure 6 have an improvement at least as large over the single-agent S2P result (8.8% for 1k samples, and 4.1% for 10k samples), than single-agent S2P has over the supervised learning baseline without self-play (7.8% for 1k samples, and 4.4% for 10k samples). Also, population-based methods help with some parts of our analysis (for example, generating Figure 4c: the ‘perfect emcomm baseline’ wouldn’t be as intuitive without having the distiller trained on a population of such languages — see our response to Reviewer #2). With this being said, we agree that the presentation of this result could be changed in our paper to de-emphasize it, and will work on this in the final version. We’d love to hear if the reviewer has recommendations for this.
>
> > Test on more variations of data size for better visualization
>
> We did run experiments with 2k and 5k samples and show the training curves for different S2P methods in the Appendix. Due to space constraints, we chose to show results on only 1k and 10k in the main text of the paper. However, we can go ahead and add another graph showing performance vs. # samples to the final version of the paper.
>
> > Figures and captions need to be improved
>
> We thank the reviewer for these observations, and will make the changes in our final version.
>
>
> References:
> [1] Multi-agent communication and the emergence of (natural) language, Lazaridou et al, 2016.
> [2] Compositional obverter communication learning from raw visual input, Choi et al, 2018.
> [3] Emergent communication in a multi-step, multi-modal referential game, Evtimova et al, 2017.

---

### Public Comment · ~Yuchen_Lu1 · 2019-11-04
**About Implementation of the OR Game**

Dear Authors,

Thanks for the interesting work. I am currently trying to build upon your Obect Reconstruction settings. I realize that you mentioned the target language is in arbitaray order. E.g. "blue triangle shaded" would be the same as "blue shaded triangle". My question is
1. During pretraining/supervised learning, is speaker/listener trained on different permutation of the language
2. During selfplay, will the communication channel send the permuted message from speaker to listener?

---

> ### Author Response · Authors · 2019-11-11
> **Answers re: OR game**
>
> Thanks for your interest in our work!
>
> 1) By assigning each object a unique word, we simulate a perfectly compositional language. One population of speaker and listener are trained on a fixed order of words. Different populations are trained on different permutations of the language. So there is no varying permutation during training.
>
> 2) During self-play, the message received by the listener is the same as the one sent by the sender.
>
> Hope this helps!

---

### Decision · Program_Chairs · 2019-12-19

**Decision:**

Accept (Poster)

**Comment:**

This paper investigates how two means of learning natural language - supervised learning from labeled data and reward-maximizing self-play - can be combined. The paper empirically investigates this question, showing in two grounded visual language games that supervision followed by self-play works better than the reverse.

The reviewers found this paper interesting and well executed, though not especially novel. The last is a reasonable criticism but in this case I think a little beside the point. In any case, since all the reviewers are in agreement I recommend acceptance.